# Electromagnetic wave-based extreme deep learning with nonlinear time-Floquet entanglement

Ali Momeni[1] & Romain Fleury [1✉]

Wave-based analog signal processing holds the promise of extremely fast, on-the-fly, power-efficient data processing, occurring as a wave propagates through an artificially engineered medium. Yet, due to the fundamentally weak non-linearities of traditional electromagnetic materials, such analog processors have been so far largely confined to simple linear projections such as image edge detection or matrix multiplications. Complex neuromorphic computing tasks, which inherently require strong non-linearities, have so far remained out-of-reach of wave-based solutions, with a few attempts that implemented non-linearities on the digital front, or used weak and inflexible non-linear sensors, restraining the learning performance. Here, we tackle this issue by demonstrating the relevance of time-Floquet physics to induce a strong non-linear entanglement between signal inputs at different frequencies, enabling a power-efficient and versatile wave platform for analog extreme deep learning involving a single, uniformly modulated dielectric layer and a scattering medium. We prove the efficiency of the method for extreme learning machines and reservoir computing to solve a range of challenging learning tasks, from forecasting chaotic time series to the simultaneous classification of distinct datasets. Our results open the way for optical wave-based machine learning with high energy efficiency, speed and scalability.

---

[1] Laboratory of Wave Engineering, School of Electrical Engineering, Swiss Federal Institute of Technology in Lausanne (EPFL), Lausanne, Switzerland.
✉email: romain.fleury@epfl.ch

 1

Recently, artificial intelligence (AI) systems based on advanced machine learning algorithms have attracted a surge of interest for their potential applications in processing the information hidden in large datasets[1,2]. Wave-based analog implementations of these schemes, exploiting microwave or optical neural networks, promise to revolutionize our ability to perform a large variety of challenging data processing tasks by allowing for power-efficient and fast neuromorphic computing at the speed of light. Indeed, wave-based analog processors work directly in the native domain of an analog signal, processing it while the wave propagates through an engineered artificial structure (metamaterials and metasurfaces)[3–6], as previously established in the cases of simple linear operations such as image differentiation, signal integration, and integro-differential equations solving[7–17]. For more complex processing tasks, for example, image recognition or speech processing, both nonlinearity and a high degree of interconnection between the elements are desired, requirements that have led to various proposals of neuromorphic processors exploiting optical diffraction, coupled waveguide networks, disordered structures[18–27], or coupled oscillator chains[28,29]. A particularly vexing challenge, however, is the implementation of nonlinear processing elements. While power-efficient neuromorphic schemes require a pronounced, particular form of nonlinearities, optical non-linearities, such as in Kerr dielectrics, are typically weak at low intensities, and cannot be much controlled. This leads to sub-optimal systems that must operate with high input powers[25,30–32]. As an alternative, non-linearities that are external to the wave-based processor have also been considered, for example by exploiting the intensity dependency of a sensor, that needs an additional electronic interconnection. Unfortunately, exploiting such weak and non-controllable non-linearities drastically confines the performance of most machine learning schemes, and the relevance of wave-based platforms has so far been largely restricted to the implementation of simple linear matrix projections.

Here, we propose to leverage the physics of wave systems that are periodically modulated in time, the so-called time-Floquet systems[33–48], to solve this vexing challenge by implementing a strong, controllable nonlinear entanglement between all the neuron signals. We propose to use a simple, thin, uniform dielectric slab, whose refractive index is slowly and weakly modulated in time. With the addition of linear random scattering disorder, we implement very efficient recurrent neural networks (RNNs) schemes, namely extreme learning machine (ELM) and reservoir computing (RC). We demonstrate the high accuracy of our Floquet extreme learning machine in challenging computing tasks, from the processing of one-dimensional data (learning nonlinear functions), to challenging multi-dimensional data (e.g., the abalone dataset classification problem). We also demonstrate the flexibility of our scheme that can be multiplexed to tackle two unrelated classification tasks at the same time, simultaneously sorting COVID-19 X-ray lung images and handwritten digits. Finally, we validate our Floquet RC by predicting the time evolution of a chaotic system over a large time period (the Mackey-Glass time-series). The reservoir size of the proposed wave-based reservoir computing system is enhanced by leveraging both spatial and spectral domains in order to improve the learning performance compared to prior works, without imposing additional filters or a larger computational overhead. Such extreme time-Floquet analog learning machines are not only fast, easy-to-train, power-efficient, and versatile, but also feature a unique accuracy performance that is comparable to that obtained with the best digital schemes.

## Results

We consider a particular class of neural networks, known as recurrent neural networks (RNNs). RNNs are ideal to process intricate data due to the internal cyclic connections between internal neurons, whose outputs depend on both the current inputs and the previous states of the neurons[49]. This memory effect allows RNNs to detect recursive relations in the data, which are relevant for example to process temporal signals. In digital implementations, however, the heavy internal connectivity matrices that are involved in the training process make RNNs particularly computationally expensive and complicated[50–53]. In order to solve these challenges, a number of alternative computing approaches such as long short-term memory (LSTM)[54], echo state networks (ESNs)[55], extreme learning machines (ELMs)[56–58], and reservoir computing (RC)[51–53,59] have emerged. These schemes are particularly well suited for wave-based implementations, because wave propagation inherently relies on the inertial memory of the medium, which can be enhanced and engineered by leveraging resonant elements, or multiple scattering. In addition, wave interferences are a particularly efficient way to create a high degree of interconnections between a large set of inputs.

Our time-Floquet neuromorphic processor implements an ELM, schematically shown in Fig. 1a. ELMs, or closely related methods based on random neural networks[60] or support vector machines[61], are a powerful scheme in which only a last layer of connections is trained (in blue). The fundamental mechanism is the use of the non-trained part of the network, whose layers are represented in gray and red in Fig. 1a, in order to establish a nonlinear mapping between the initial space of the dataset and higher-dimensional feature space, where a properly trained classifier performs the separation and classification. In our case, this nonlinear mapping is performed by letting one of the non-trained layers (in red) be weakly modulated in time at a frequency much lower than the one of the signal, and with a modulation phase that depends on the input state.

A concrete implementation of this scheme in a wave platform is shown in Fig. 1b. It consists of three parts: (i), an array of monopole antennas that radiates the various components of the input vector into the surrounding medium; (ii), a propagation space composed of a few scatterers and a thin dielectric slab, called a scattering time-modulated slab (STMS), whose index of refraction is weakly modulated in time; and (iii), the output layer made of an array of receiving antennas and a single dense layer, digitally trained to perform the desired regression or classification tasks. At the input layer, the input vector $\zeta^{\text{in}}$ with components $\zeta_1^{\text{in}}, ..., \zeta_N^{\text{in}}$ is first encoded into $N$ signals $s_i^{\text{in}}$, injected directly into the source antenna array (see further details in section Details of the proposed wave-based ELM architecture of the Methods). We assume that $\zeta^{\text{in}}$ is modulated at two distinct close-by frequencies $\omega_1$ and $\omega_2$, such that:

$$s_i^{\text{in}} = \zeta_i^{\text{in}}\big(\sin(\omega_1 t) + \sin(\omega_2 t)\big). \tag{1}$$

The permittivity $\epsilon_r$ of the STMS is modulated with a depth $\delta_m$ and a phase $\phi$, at a frequency $\omega_m = |\omega_1 - \omega_2|/2$, so that $\epsilon_r = \epsilon_s + \delta_m \cos(\omega_m t + \phi)$. This choice of modulation frequency allows for the two input frequencies to be efficiently mixed at the dominant Floquet harmonic $(\omega_1 + \omega_2)/2$ (see Fig. 1c). As we will now see, the reflection and transmission coefficients of Floquet Harmonics can show a strongly nonlinear dependency on the modulation phase, a key property that we will leverage to make the ELM very efficient.

To understand how time-Floquet systems can be used to induce large nonlinear entanglement between the incident and reflected signals, let us consider the toy model of a generic two-port time-Floquet system, where incident and reflected signals at ports 1 and 2 are represented by their time-varying complex amplitudes $a_{1,2}(t)$ and $b_{1,2}(t)$. This model applies for each plane wave incident on our STMS, with transverse wave number $k$, on

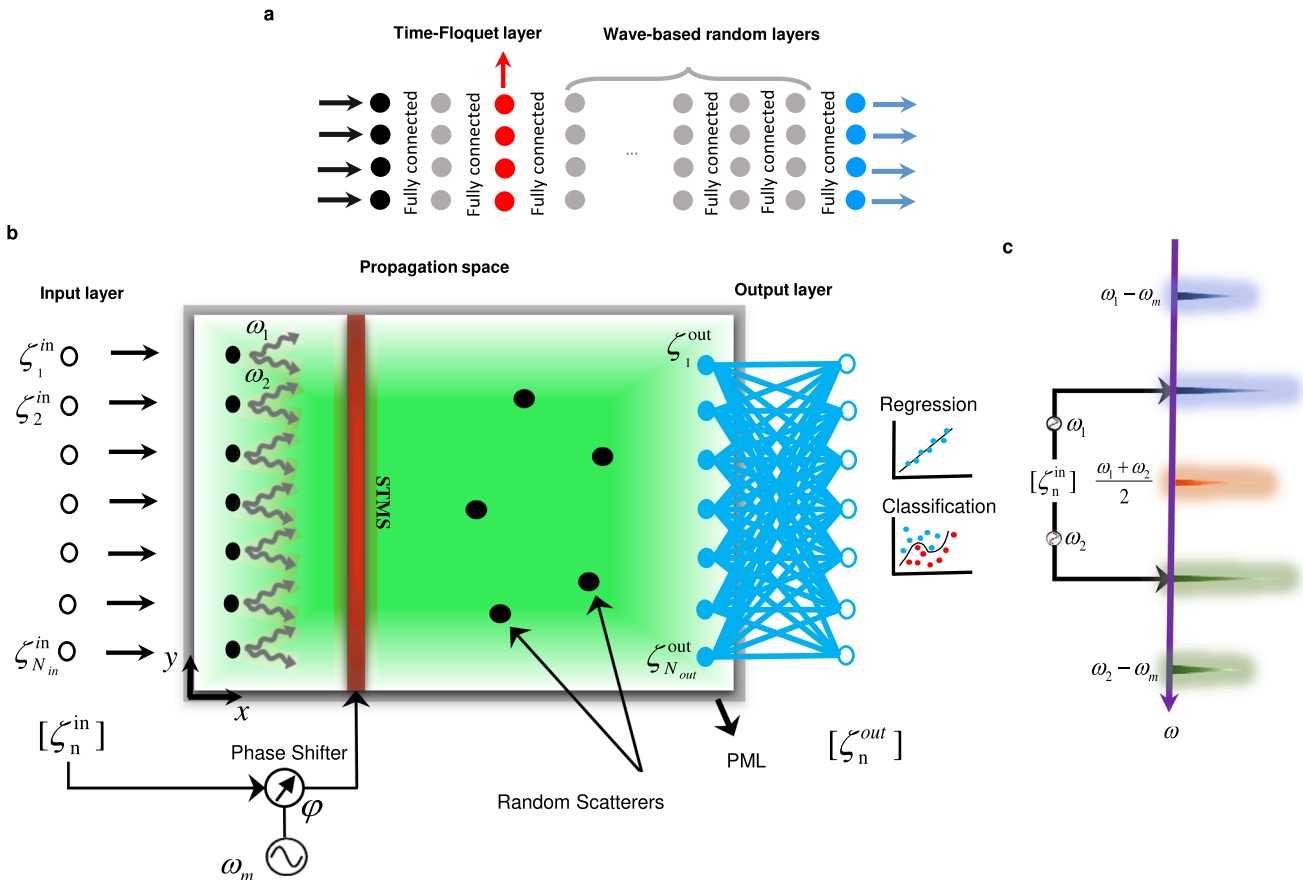

**Fig. 1 Wave-based time-Floquet extreme learning machine. a** Schematic of a neural network including a time-Floquet layer made from neurons whose properties are modulated periodically in time, and traditional random layers. Only the last layer (output) is trainable. **b** Concrete implementation with electromagnetic waves. The input signals $\zeta_n^{in}$ are modulated at $\omega_1$ and $\omega_2$. The sums of these frequency components forms input signals that are independently radiated into the surrounding space by an array of source antennas (black disks). As the waves propagates in the green region, they encounter a thin dielectric slab whose index of refraction is modulated at the frequency $\omega_m = |\omega_1 - \omega_2|/2$, as well as five sub-wavelength scatterers, randomly located in the domain. The modulation phase depends on the input vector $\zeta_n^{in}(t)$. The gray rectangle represents an absorbing boundary layer. The outputs $\zeta_n^{out}(t)$ are fed into an adaptable blue dense layer, and used for regression and classification. **c** Nonlinear phase entanglement. The modulated slab mixes signals at $\omega_1$ and $\omega_2$ into Floquet harmonics spaced by $\omega_m$, whose interferences depend non-linearly on the input vector.

which the actual field can be decomposed. Assuming the modulation frequency $\omega_m$ to be much smaller than the operation frequency $\omega_k$[62,63], we can neglect dispersive effects and write the following instantaneous relation between the signals at each ports[62–64]:

$$\begin{bmatrix} a_1(t) \\ b_1(t) \end{bmatrix} = \tilde{\Psi}(\omega_k, t) \begin{bmatrix} a_2(t) \\ b_2(t) \end{bmatrix}, \quad (2)$$

where $\tilde{\Psi}(\omega_k, t)$ is the transfer matrix at $\omega_k$, which varies slowly with time. Taking the Fourier transform of both sides yields

$$\begin{bmatrix} A_1(\omega) \\ B_1(\omega) \end{bmatrix} = \tilde{\Psi}(\omega_k, \omega) * \begin{bmatrix} A_2(\omega) \\ B_2(\omega) \end{bmatrix}$$
$$= \int \tilde{\Psi}(\omega_k, \omega - \omega') \begin{bmatrix} A_2(\omega') \\ B_2(\omega') \end{bmatrix} d\omega', \quad (3)$$

Since the scattering process into each Floquet harmonic component is linear, we can define the reflection and transmission coefficients into each harmonic as $R_0(\omega_k + n\omega_m) = B_1(\omega_k + n\omega_m)/A_1(\omega_k)$ and $T_0(\omega_k + n\omega_m) = A_2(\omega_k + n\omega_m)/A_1(\omega_k)$. A direct calculation shows that (see Sec. 1 of the Supplementary Material for detail derivations):

$$R_\phi(\omega_k + n\omega_m) = e^{in\phi} R_0(\omega_k + n\omega_m) \quad (4)$$

$$T_\phi(\omega_k + n\omega_m) = e^{in\phi} T_0(\omega_k + n\omega_m), \quad (5)$$

where we have used the notation $R_\phi$ to highlight the dependency of the scattering coefficients on the modulation phase $\phi$. These equations imply that upon adding a phase delay $\phi$ to the modulation, the generated frequency harmonic of order $n$ will acquire a phase shift of $n\phi$, both for the forward and backward scattered plane waves. On the other hand, the amplitude of harmonic waves is constant when we alter the phase delay.

Now, consider the superposition of two incident plane waves at frequencies $\omega_1$ and $\omega_2$. Recalling our choice of modulation frequency, namely $\omega_m = |\omega_1 - \omega_2|/2$, we can write the reflection and transmission waves for all Floquet harmonic components of frequency $\omega_1 + n\omega_m = \omega_2 + m\omega_m$ by using the superposition principle:

$$|R_\phi'| = |e^{in\phi} R_0(\omega_1, \omega_1 + n\omega_m) + e^{im\phi} R_0(\omega_2, \omega_2 + m\omega_m)| \quad (6)$$

$$|T_\phi'| = |e^{in\phi} T_0(\omega_1, \omega_1 + n\omega_m) + e^{im\phi} T_0(\omega_2, \omega_2 + m\omega_m)|, \quad (7)$$

where $n$ and $m$ are the orders of the Floquet harmonics with respect to $\omega_1$ and $\omega_2$, respectively. A particular example is the harmonic located at the average frequency $\omega = (\omega_1 + \omega_2)/2$, for which $n = 1 = -m$ (orange spectrum in Fig. 1c). According to Eqs. 6 and 7, the relation between the modulation phase and the

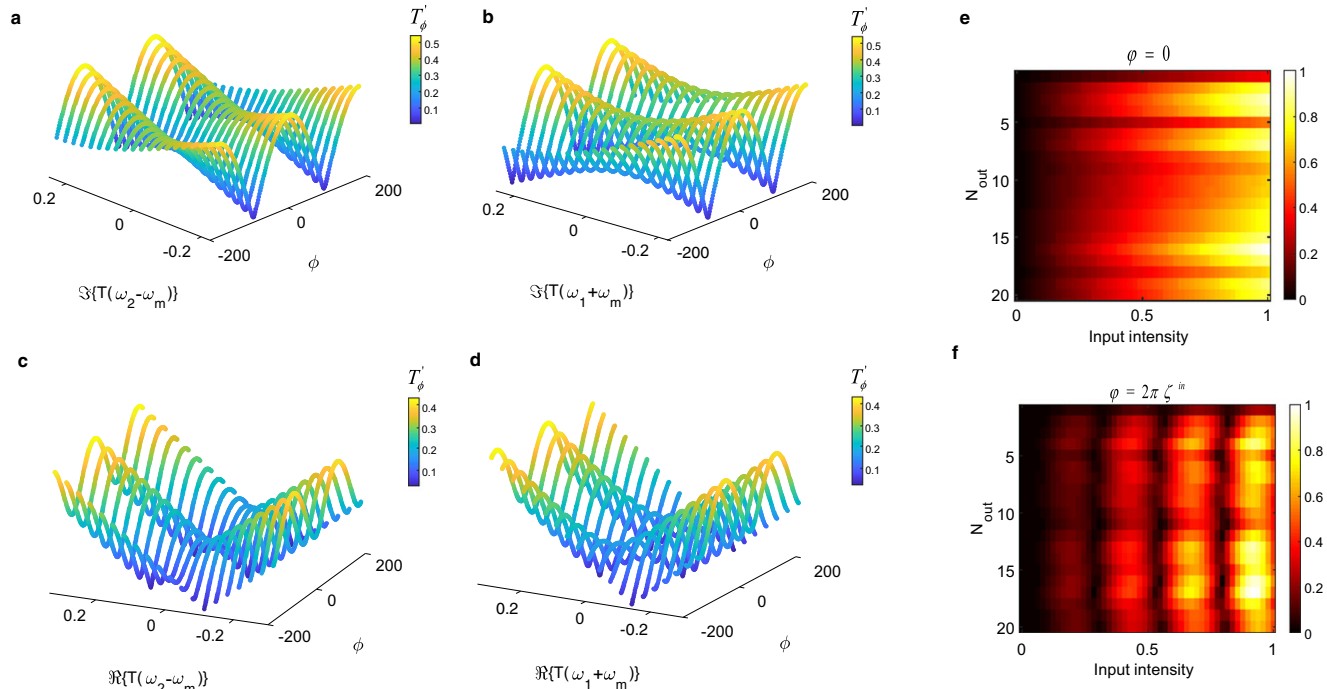

**Fig. 2 Nonlinear Floquet entanglement. a–d** Theoretical demonstration of the nonlinear dependency of the intensity $T'_\phi$ of the central Floquet harmonic ($\omega = (\omega_1 + \omega_2)/2$) on both the modulation phase $\phi$ and the real or imaginary part of one of the generated harmonics. The results are based on Eq. 7. The fixed parameters for **a–d** are: **a** $T(\omega_1 + \omega_m) = 0.1 - 0.25i$ and $\Re\{T(\omega_2 - \omega_m)\} = 0.1$. **b** $T(\omega_2 - \omega_m) = 0.1 - 0.25i$ and $\Re\{T(\omega_1 + \omega_m)\} = 0.1$. **c** $T(\omega_1 + \omega_m) = 0.1 - 0.05i$ and $\Im\{T(\omega_2 - \omega_m)\} = 0.05$. **d** $T(\omega_2 - \omega_m) = 0.1 - 0.05i$ and $\Im\{T(\omega_1 + \omega_m)\} = 0.05$. **e, f** The numerical demonstration of linear/nonlinear Floquet entanglement for the central harmonic wave for different readout nodes in terms of input intensity, for static (**e**) and dynamic phase delays (**f**).

intensity of scattered harmonic fields is highly nonlinear. In fact, we can control the amplitude of the Floquet harmonics only by changing the modulation phase. In order to have a nonlinear input–output mapping, we must therefore entangle the phase delay with the input vector (i.e., $\phi = f(\zeta^{in})$). This can be done by using a simple voltage-controlled phase shifter (VCP) (see further details in section Details of the proposed wave-based ELM architecture of the Methods). In other words, the value of the modulation phase is directly determined by the value of the input vector, which is fixed when the system is excited, automatically turning the scattering process into a highly nonlinear function of the input, regardless of the input power. This makes such time-Floquet nonlinear entanglement highly advantageous in neuromorphic computing schemes.

To exemplify the strong nonlinear response of the proposed system, we plot the amplitude of the transmitted central harmonic ($\omega = (\omega_1 + \omega_2)/2$) as a function of various variables, including the phase delay $\phi$. The results are displayed in Fig. 2a–d. We fix one of the harmonics and plot $T'_\phi$ versus the modulation phase and the real or imaginary part of the other transmitted harmonics, $T(\omega_1 + \omega_m)$ (or $T(\omega_2 - \omega_m)$). As we can see in Fig. 2a–d, we indeed obtain a complex nonlinear semi-sinusoidal form for $T'_\phi$, upon altering the modulation phase. The dependency on the real or imaginary parts of the other transmitted harmonic is also always nonlinear.

Next, we implement the entanglement with the input vector to demonstrate the complex nonlinear behavior of the Floquet system, using a full-wave finite-difference time-domain simulation of the setup of Fig. 1b (see Methods). We compute the intensity of the central harmonic with respect to the input intensity for two different scenarios: a static phase delay and an entangled phase delay. In the first scenario, the phase delay is fixed and not dependent on the input ($\phi = 0$), and as shown in Fig. 2e, the

harmonic intensities are linear in terms of input intensities. In the second scenario, the delay phase is a simple linear function of the input (i.e., $\phi = 2\pi\zeta^{in}$). Figure 2f shows the complex nonlinear form of the proposed system. The oscillating nonlinear mapping performed by the proposed system is completely different from any earlier approach. As we will show, it is surprisingly effective in transforming the input data space to a nearly linearly separable output data space.

Note that another alternative approach to reach such a highly nonlinear input–output mapping is to entangle the input data with the modulation depth instead of the modulation phase. In this case, no phase shifters are needed. In section 2 of the Supplementary Material, more explanations about this alternative can be found, including a demonstration of its high performance in terms of transforming the input data space to a nearly linearly separable output data space.

**Learning highly nonlinear functions**. We now demonstrate the performance of the Floquet ELM by starting with simple regression problems, on a dataset generated with nonlinear relations. Such a dataset is often used as a standard benchmark in machine learning since linear regression of a nonlinear function is impossible without a nonlinear transformation[31,56]. The input information ($\zeta^{in}$) is a set of randomly generated numbers between $-\pi$ to $\pi$ and the corresponding output labels ($y_i$) are generated according to nonlinear functions, namely $y_1 = \alpha \sin(4\pi\zeta^{in})(|\zeta^{in}|/\pi)$, $y_2 = \text{rect}(\zeta^{in})$ (pulse function), and $y_3 = \sin(\pi\zeta^{in})/(\pi\zeta^{in})$. We use 1000 randomly generated samples, which lie in $[-\pi, \pi]$ to cover the entire characteristic behavior of the function. We map each input value to a vector by multiplying it with a fixed random 1D vector (mask), here of dimension $1 \times 10$. In this task, we use 10 and 20 input and readout nodes, respectively. By recording the intensity of the harmonics in

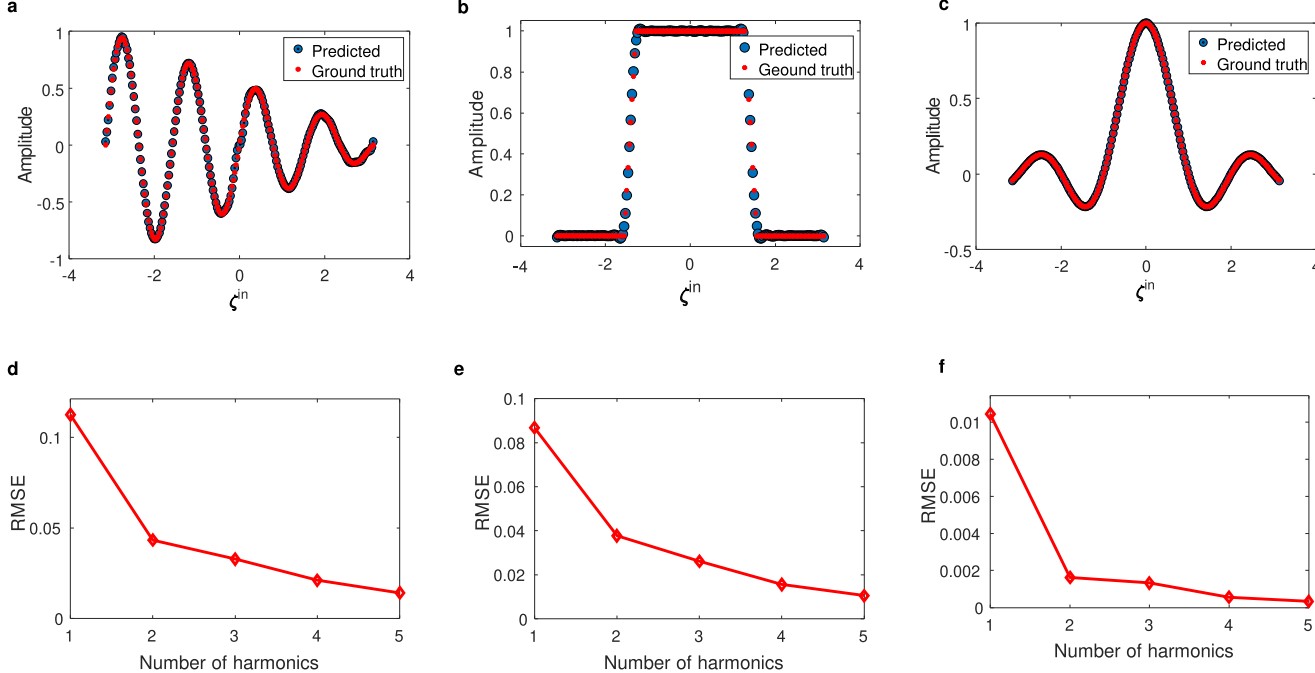

**Fig. 3 Floquet extreme learning for highly nonlinear maps. a–c** Comparison between ground truth and predicted values for three different nonlinear functions: $y_1 = \alpha \sin(4\pi\zeta^{in})(|\zeta^{in}|/\pi)$, and $y_2 = \mathrm{rect}(\zeta^{in})$ and $y_3 = \sin(\pi\zeta^{in})/(\pi\zeta^{in})$, respectively. **d-f** The corresponding values of root-mean-square error (RMSE) upon increasing the numbers of involved Floquet harmonics at the readout nodes.

the readout nodes of many input values, linear regression is performed on the output data (see Fig. 3a–c). A remarkable learning performance, with very low root-mean-squared error (RMSE) for all three nonlinear functions, is obtained. Interestingly, in the proposed wave-based neural network with a nonlinear time-Floquet layer, the multiple generated harmonic fields can be used to extend the dimension of the nonlinear mapping, and increasing their number improves the accuracy of classification/regression. This tendency is demonstrated in Fig. 3d–f), which plots the RMSE versus the number of considered Floquet harmonics. This mechanism is a clear advantage of the Floquet ELM: by involving a higher number of scattered harmonics, we can improve the RMSE and enhance the accuracy of learning with no additional computational cost. It should be noted that in order to compute outputs at the decision layer, we simply rescale the linear regression weights without having to use additional filters. (see further details in section Training of readout of the Methods).

We can explain the learning principle of the proposed computing system with well-known kernel methods. Kernel methods use kernels (or basis functions) to map the input data into a feature space. After this mapping, simple models can be trained on the new feature space, instead of the input space, which can result in an increase in the performance of the models[65,66]. We can describe the projections of input samples in the feature space by $T' = H_{non}(p(\zeta^{in}))$, where $p$ is encoding function, here for example $p(\zeta^{in}) = \zeta^{in}(\sin(\omega_1 t) + \sin(\omega_2 t))$ and $H_{non}$ is a nonlinear and complex function associated with the time-Floquet entanglement. Essentially, $H_{non}$ can be seen as an explicit form of optical kernel function which contains both the multiple scattering occurring in the media and the complex nonlinearity form. This kernel contains several polynomial basis functions, $\{x, x^2, x^3, x^4, \dots\}$ (we can theoretically show it by Taylor expansion of Eqs. 6 and 7). Hence, we have a combination of different orders of polynomial mappings with random coefficients in the feature space for each readout node, resulting in transferring different features of the input data into the feature

space. The proposed kernel is thus expected to be very efficient in performing all tasks, even when compared with a strongly nonlinear Kernel such as the modulus square operator ($x^2$), typically found in sensors and detectors used in prior arts.

To prove this quantitatively, we compare the feature space projection of our kernel with the form of nonlinearity that is most commonly used: a square-law at the detector, $x^2$. As a reference, we also look at the purely linear case. As an example, we consider again the nonlinear interpolation of $y = sinc(x)$. In order to perform well, the data projected in the feature space should be highly nonlinear with respect to the feature coordinates. To visualize this, we use principal component analysis (PCA) to reduce the dimension of the output data, because it lives in a high-dimensional space (10 dimensions, set by the number of readout nodes). PCA is a kind of linear projection that consists in transforming correlated variables into new variables, decorrelated from each other. These new variables are called "principal components" or principal axes[67]. In Fig. 4a, b, we calculate and plot this projected data in a 3D PCA space for three distinct cases: linear, $x^2$ nonlinearity, and time-Floquet entanglement. Panels (a) and (b) show that in both the linear and $x^2$ cases, the projected data is on a line, whereas in the case of the time-Floquet entanglement, the data follows a highly nonlinear relation with respect to the principal axes. For this reason, the $sinc(x)$ interpolation fails when using both a linear system or one with $x^2$ nonlinearity at the detector (see Fig. 4c, d). Conversely, time-Floquet entanglement is extremely good at performing the sinc problem (see Fig. 4e).

**Abalone dataset.** In the previous section, we have used our Floquet ELM to learn nonlinear functions and their interpolation capability. However, interpolation is not always the relevant task, especially in complex inference problems. Therefore, we now move to a more challenging multivariable problem: the abalone dataset. This dataset is one of the most used benchmarks for machine learning and concerns the classification of sea snails in

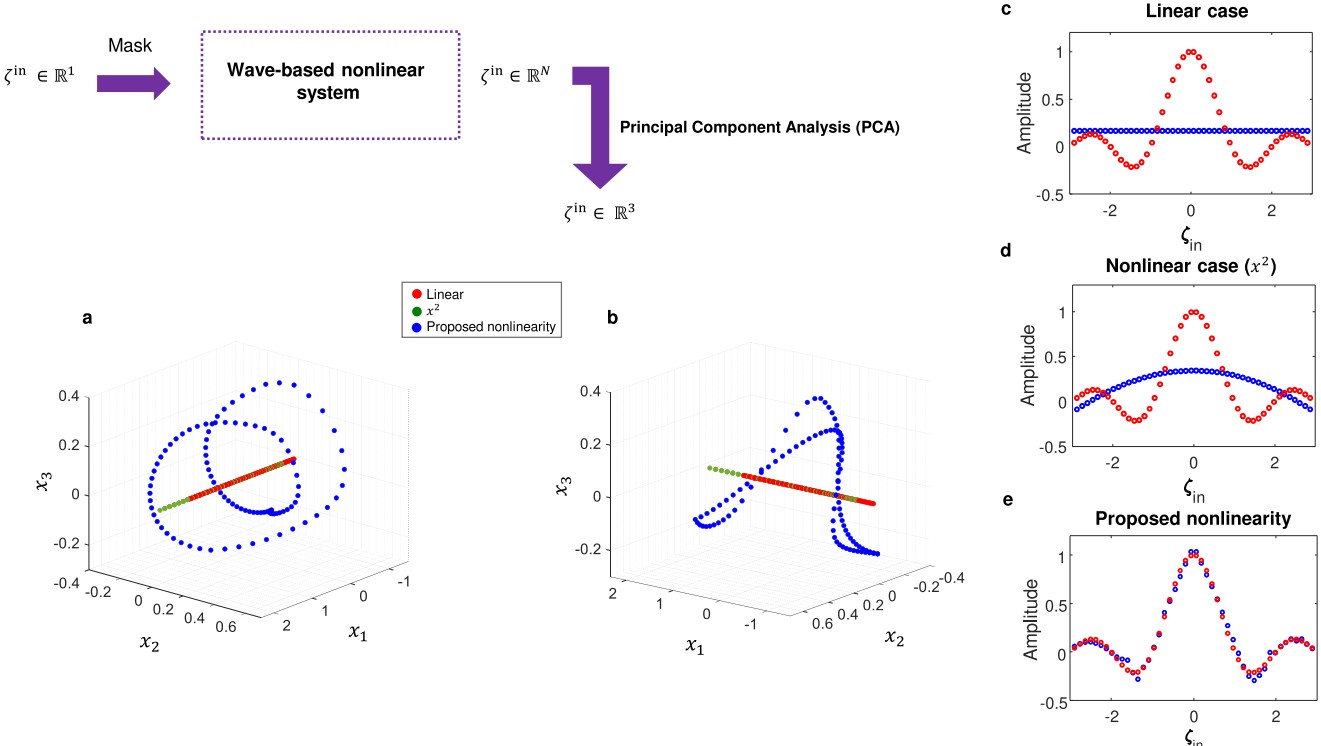

**Fig. 4 PCA analysis of the proposed optical kernel. a, b** Two different perspectives of Projected data for three different cases: (i) linear case, (ii) square-law nonlinearity ($x^2$), and (iii) the proposed nonlinearity. **c–e** $sinc(\zeta^{in})$ interpolation results for three aforementioned cases, respectively.

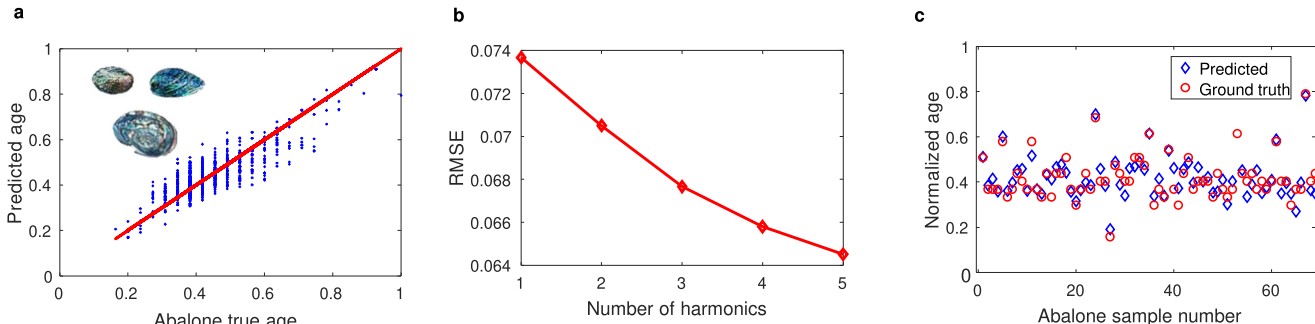

**Fig. 5 Floquet extreme learning for multivariable regression. a, b** Learning of the abalone dataset and corresponding RMSE for different numbers of considered harmonic waves, respectively. **c** Comparison between predicted data (blue) and ground truth (red).

terms of age and physical parameters. It lists eight physical features of sea snails that can be used for the prediction of their age. To tackle this problem with our Floquet ELM, we encode the 8 physical features of sea snails on our input nodes (8 input nodes), and consider 50 readout nodes to feed the decision layer, which performs a linear regression. Figure 5a presents the true ages and the corresponding predictions; the figure indicates that the framework learns the ages of the abalone with remarkable accuracy. For a direct comparison, we plot the predicted values for 75 random input data (Fig. 5c). The RMSE with respect to a number of harmonic waves is plotted in Fig. 5b. A remarkable accuracy (RMSE = 0.064) can be achieved by considering five generated harmonics. The achieved RMSE is smaller than the best value reported in prior art[31].

**Parallel image classifications**. Another remarkable feature of time-Floquet systems is that since the inputs are modulated at a certain carrier frequency, we can use several frequency bands and

multiplex different signals to classify them simultaneously using the same system, and at no additional cost in terms of power consumption. Let us now demonstrate this in a specific complex parallel classification task. We examine the possibility to perform parallel image classification using two wavelength inputs. We use two distinct datasets: the MNIST dataset of handwritten digits and the COVID-19 X-ray images (see Fig. 6a, b). We resize all of the images into $10 \times 10$ pixels, down-sampling them to decrease the number of input and readout nodes and the total size of our structure. In this task, we use 100 nodes to encode the images with the amplitude of the input waves, and 100 readout nodes. The MNIST data are encoded onto a (randomly selected) frequency range from 4 to 4.125 THz, and the COVID-19 data are encoded between 4.375 and 4.5 THz (see the red and blue frequency bands in Fig. 6c). In the output layer, we use Softmax regression to perform classifications (See Methods).

The training results are shown in Fig. 6d–g. The observed test accuracies were 88.2% for the COVID-19 and 85.3% for the MNIST datasets. These classification accuracies are competitive.

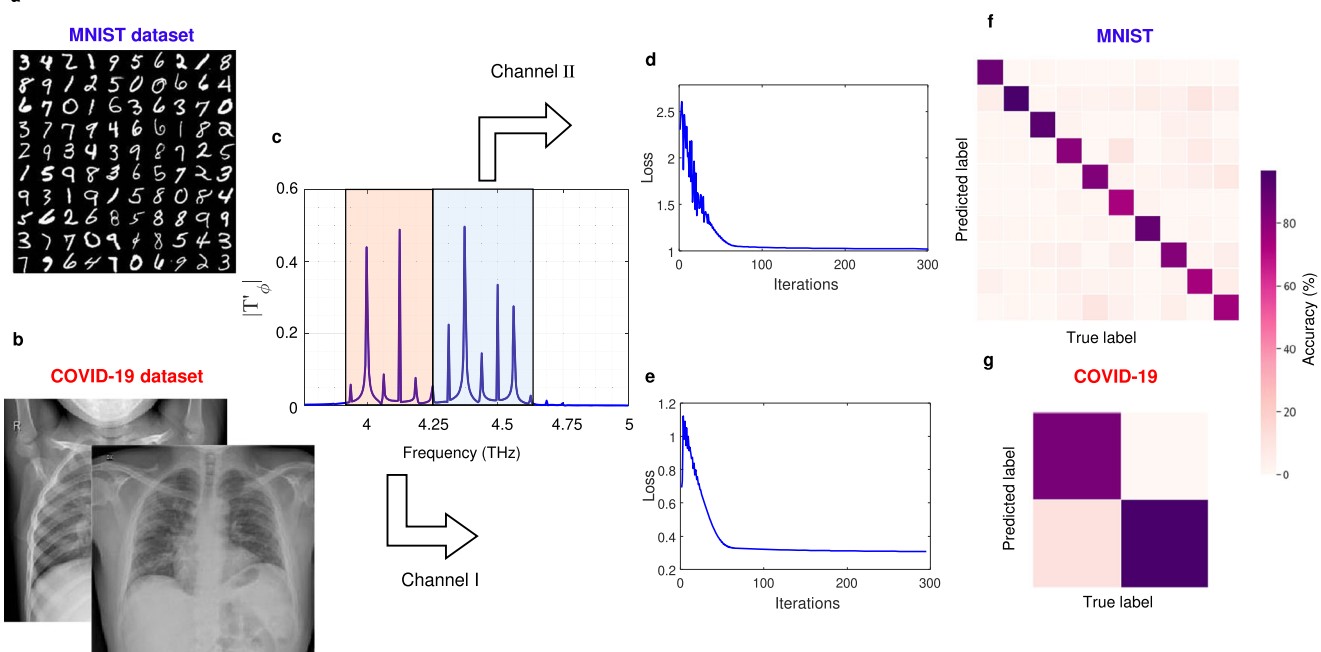

**Fig. 6 Floquet extreme learning for parallel image classification. a, b** Examples of realizations taken from the MNIST and COVID-19 X-ray datasets. **c** Simulated spectra of readout nodes for two different channels. **d, e** Evolution of the loss function for MNIST and COVID-19 classifications, respectively. **f, g** Corresponding confusion matrices, for condition classification.

For example, they are higher than the ones reported in reference[50] (parallel image classification). Also, the classification-accuracy results are comparable with other relevant works despite decreasing the pixel sizes of all images[53]. In addition, this frequency multiplexing technique is the first demonstration of wave-based parallel task processing with extreme deep learning. This enables the use of wide bandwidth as a computational resource, which significantly boosts computation efficiency.

**Nonlinear Time-Floquet-based RC system for autonomous forecasting chaotic time-series.** To show the high versatility of the proposed nonlinear time-Floquet neuromorphic computing system, we slightly modify it to implement a reservoir computing (RC) scheme. Consider an input vector $i(t)$ that is injected into a high-dimensional dynamical system called the reservoir. The reservoir is described by a vector $h(t)$ and the initial state of the reservoir is defined randomly. Let the $W_{res}$ matrix define the internal connections of the reservoir nodes and the $W_{in}$ matrix define the connections between the input and the reservoir nodes. Both matrices are initialized randomly and fixed during the whole RC training process. The state of each reservoir node is a scalar $h(t)$, which evolves according to the following recursive relation:

$$h(t + \tau) = F\big(w_{\text{in}}i(t) + w_{\text{res}}h(t)\big) \tag{8}$$

where $\tau$ is the discrete time-step of the input and $F$ is a nonlinear function. From Eq. 8, we see that the reservoir is defined as a dynamical system provided with a unique memory property; namely, each consequent state of the reservoir contains some information about its previous states and about the inputs injected until that time. In the training phase, the input $i(t)$ is fed to the reservoir, and the corresponding reservoir states are recursively calculated. The final step of the information processing is to perform a simple linear regression in order to minimize the RMSE that adjusts the $W_{out}$ weights. The output can be computed with $O(t) = W_{out}h(t)$. It should be noted that the output weights are the only parameters that are modified during

the training. The input and reservoir weights are fixed throughout the whole computational process, and they are used to randomly project the input into a high-dimensional space, which increases the linear separability of inputs.

In our concrete scheme, we implement this memory using a feedback loop, and use the intensity of harmonic waves as reservoir states. The reservoir computing in our scheme can be described by the following recursive relation:

$$T'_\phi(t + \tau) = F\Big(w_{\text{in}}i(t) + w_{\text{res}}v_h T'_\phi(t)\Big) \tag{9}$$

where F the nonlinear function describing our system, $v_h$ is a tunable parameter that selects one (or more) harmonics as reservoir states, and $T'_\phi$ is the intensity of transmission harmonic waves. In general, the RC and its different implementations have proven to be very successful for various tasks, such as spoken digits recognition, temporal Exclusive OR task, Mackey-Glass, or Nonlinear Autoregressive Moving Average time-series prediction[68,69].

We use the nonlinear time-Floquet RC for the prediction of chaotic time-series. Forecasting chaotic time-series is an extremely difficult task due to the accumulation of quantitative differences between the ground truth and the predicted value in subsequent predictions, which lead to exponential errors at large times. Indeed, the positive Lyapunov exponent in chaotic systems leads to exponential growth for the separation of close trajectories, so that even small errors in prediction can quickly lead to divergence of the prediction from the ground truth[49]. We test our system using the Mackey-Glass time-series defined by[49,70].

$$\frac{dy}{dt} = \beta \frac{y(t - \tau)}{1 + \big(y(t - \tau)\big)^n} - \gamma y(t) \tag{10}$$

Unlike deterministic equations, predicting such time-series for specific values of parameters is difficult and thus has been widely used as a benchmark for challenging forecasting tasks. To obtain chaotic dynamics, here, we set the parameters $\beta = 0.2$, $\gamma = 0.1$,

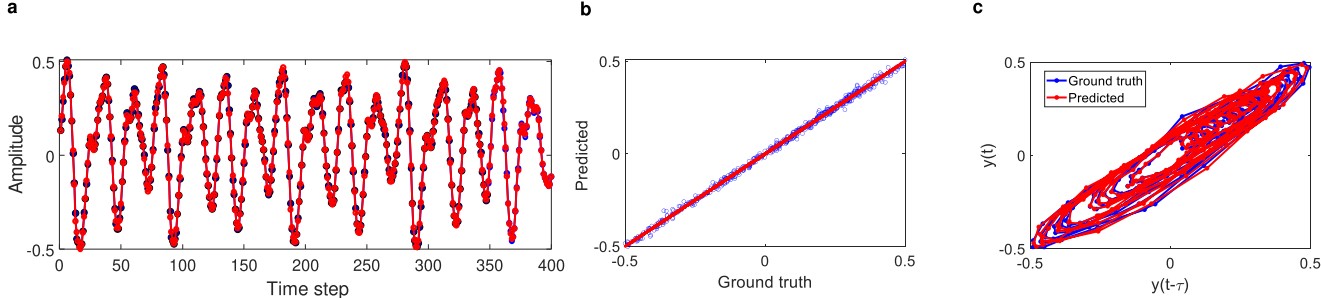

**Fig. 7 Floquet reservoir computing for forecasting the chaotic Mackey-Glass time-series. a** Training results: The ground truth (blue) and the predicted output from the RC system (red) are plotted. **b** Corresponding results of linear regression. **c** Trace of time-series values in phase space, with respect to the previous time-step for both ground truth and predicted values.

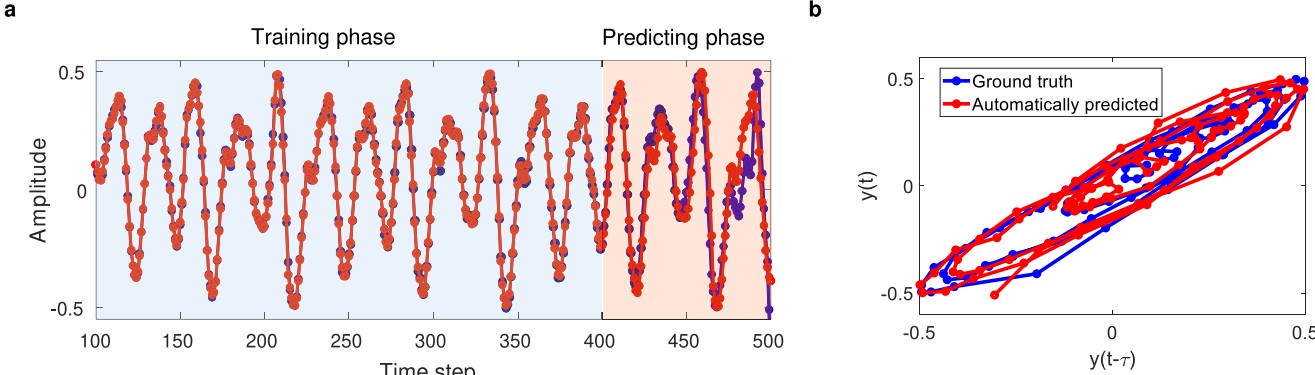

**Fig. 8 Autonomous forecasting of Mackey-Glass time-series.** Training and forcasting results: **a** The ground truth (blue) and the predicted output from the RC system (red) for 100 next time-steps are plotted. **b** Trace of time-series values, phase space, for predicting phase with respect to the previous time-step.

$\tau = 18$, $n = 10$. During the training phase, as soon as the reservoir states are calculated, a simple linear regression is executed to adjust the $W_{out}$ weights such that their linear combination with the calculated reservoir states makes the actual output as close as possible to the next time-step of the input. Finally, to automatically predict the future evolution of $i(t)$, we make a feedback loop from the output to the input by replacing the next input $i(t+1)$ with the one-step prediction $W_{out}o(t)$, as is done in conventional RC. The ability of the proposed RC system in time-series prediction is tested using a reservoir with 100 input nodes and 50 readout nodes. We consider the middle harmonic as a reservoir state and input, $\zeta_n^{in}$, to feed our RC system for each interaction (Eq. 9). All of the intensity harmonics and reservoir states are then applied to the readout layer (see Methods) to generate the predicted data for the next time-step. Figure 7 shows the results obtained during training from the simulation. Excellent agreement between the target and the predicted value can be obtained, indicating that the trained readout weights can correctly calculate the next time-step signal on the basis of the internal states of the reservoir. Further evidence of successful training can be found by examining the network performance in regression and phase space, as shown in Fig. 7b, c, where an excellent agreement can again be observed.

The network is then used to forecast the time-series autonomously. After training for 400 time-steps, the output from the readout function, that is, the predicted data for the next time-step is then connected to the reservoir as the new input, and the system autonomously produces the forecasted time-series continuously. Figure 8 shows the results for autonomous time-series prediction using the proposed RC system. Afterward, the autonomously generated output (from the 400th time-step

onwards) still matches very well the ground truth, showing the ability of the proposed RC system to autonomously forecast the chaotic system. After more than 70 time-steps of autonomous prediction, the predicted signal starts to diverge from the correct value, which is unavoidable due to the chaotic nature of the series. Increasing the size of the reservoir further, by using more nodes and using more previous states may reduce the prediction error so that the length of accurate prediction can be increased. Another solution for long-term forecasting without increasing the dimension of the system is utilizing a periodical update procedure as in ref. [49] . In section 3 of the Supplementary Material, we compared the computing performance of the proposed system with prior works for all tasks.

In conclusion, we have shown how nonlinear Floquet entanglement can be used to enable wave-based neuromorphic computing, by allowing for strong and tailored nonlinear mapping to a higher-dimensional space without involving any nonlinear material. Our nonlinear time-Floquet learning machine can process information to compute complex tasks that are traditionally only tackled by slower, sophisticated, and digital deep neural networks. In our benchmarks, the proposed computing platform performs as well as its digital counterparts. With better energy efficiency in comparison to the previous proposals and a path to high scalability, our nonlinear time-Floquet system provides a unique solution for supercomputer-level optical computation.

## Methods

**Numerical simulations**. We use a two-dimensional finite-difference time-domain (FDTD) method for all simulations[71,72]. Figure 9 shows the rectangular layout of the employed setup. We set the parameters $\epsilon_s = 3$, $\delta_m = 0.3$, $\omega_m = |\omega_1 - \omega_2|/2$.

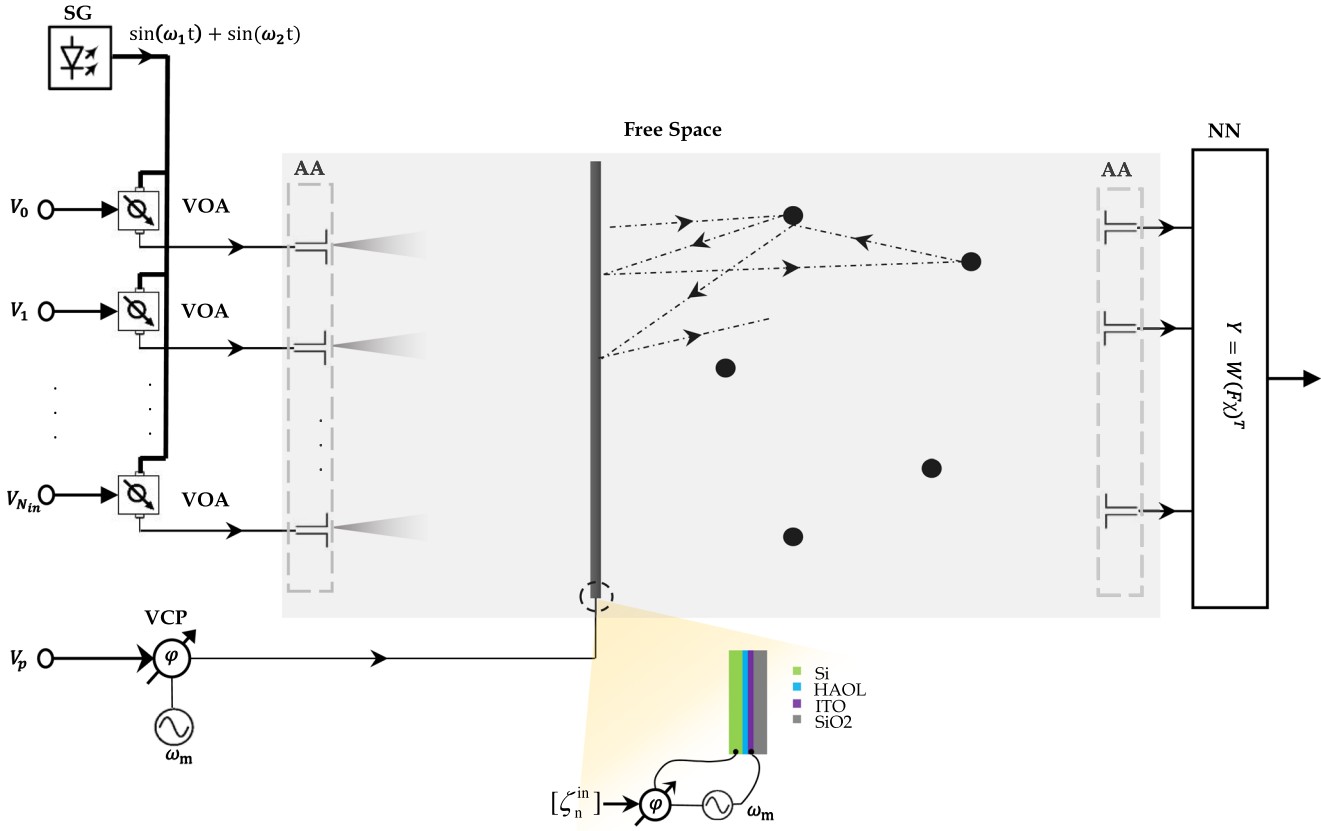

**Fig. 9 A detailed schematic of the proposed wave-based ELM architecture.** On the left, the input side includes a signal generator (SG) with variable attenuators (VOA) that encode the information. A Voltage-controlled phase shifter (VCP) is used to change the phase of the modulation depending on the voltage applied, which is fixed for each input and does not change in time. AA denotes the array of source antennas. $V_p$ is the required voltage to control the VCP ($V_p = \gamma \bar{\zeta}^{in}$), where $\bar{\zeta}^{in}$ and $\gamma$ are the mean of input vector and a scaling factor, respectively.

Furthermore, the propagation space's height and width, as well as the thickness of the STMS, are set to be $15\lambda_0$, $10\lambda_0$, and $\lambda_0/4$, respectively. We use five high index permittivity dielectrics sub-wavelength scatterers, randomly located in the propagating substrate. The time window of simulation and the spatial window (time- and space-discretization factors) is set as a $d_t = d_{x,y}/(2C)$ and $d_{x,y} = \lambda_0/30$, respectively, ($C$ is speed of light). We use 10,000 time-steps to ensure convergence.

**Details of the proposed wave-based ELM architecture**. We encode the input information with simple circuit elements, namely variable attenuators and a voltage-controlled phase shifter. These devices are set externally to a certain operating point once the input is selected, and do not change as the neural network processes a given input. Therefore, no dynamic tuning, or conversion between amplitude and phase, is needed between the temporal signal sent by the generator and the modulation signal: just like in standard learning machines, they are just set externally once an input is selected to be processed by the system. More details can be found in Fig. 9. The amplitude-coding scheme that we employ is simple and physically feasible by using variable attenuators (VOA). Since the values of input vector $\zeta_n^{in}$ are normalized between zero and one, the input information can be encoded on the amplitude of the temporal signal generated by a single signal generator (SG), as illustrated in Fig. 9. Variable attenuators provide the input node amplitudes depending on the external voltages applied. In addition, a lower frequency oscillator with a voltage-controlled phase shifter (VCP) is used to drive the modulation of the time-Floquet layer. VCPs are tunable, and the applied voltage is simply calculated by the external user from the input vector. Discussion about realistic physical platforms for realizing the time-Floquet layer is provided in section 4 of the Supplementary Material.

**Training of readout**. Here, we show how to train the decision layer using the data of temporal signals received at the readout nodes without using extra filtering operations. Consider $\zeta_q^{out}(t)$ as the temporal signal received at the output antenna $q$, and its discrete Fourier transform of the discretized signal $\bar{\zeta}^{out} = $

$\left(\zeta_q^{out}(0), \zeta_q^{out}(t_0), ..., \zeta_q^{out}((n-1)t_0)\right)^T$ defined by $y_{f_j} = \sum_{(k=0)}^{(n-1)} \zeta_q^{out}(t_k) \exp(\frac{-2\pi i}{n} jk)$

where $j = 0, ..., n-1$, $(.)^T$ is the transpose operation, and $t_0$ is the sampling time. The relation between the Fourier coefficients $y_{f_j}$ and the discretized signal is described by the so-called Vandermonde matrix:

$$\begin{pmatrix} y_{f_0} \\ y_{f_1} \\ ... \\ y_{f_{n-1}} \end{pmatrix} = \begin{pmatrix} 1 & \cdots & 1 \\ 1 & \cdots & \kappa^{n-1} \\ \vdots & \ddots & \vdots \\ 1 & \cdots & \kappa^{(n-1)^2} \end{pmatrix} \begin{pmatrix} \zeta_q^{out}(0) \\ \zeta_q^{out}(t_0) \\ ... \\ \zeta_q^{out}((n-1)t_0) \end{pmatrix} \quad (11)$$

where $\kappa = e^{\frac{-2\pi i}{n}}$. Generally, we are only interested in the middle harmonic whose component $q$ can be calculated by the scalar product $y_{f_j}^q = F_j \bar{\zeta}_q^{out}$, where $F_j = \left(1, \kappa^j, \kappa^{2j}, ..., \kappa^{(n-1)j}\right)$ and $j$ is correspond to desired harmonics. In order to train the readout function by linear regression (which is commonly defined by a linear matrix operation of the form $Y = WX^T$, where $W$ is the weights matrix), we must compose both operations, multiplying the $F_j$ and weight matrices:

$$Y = W(F_j X)^T \quad (12)$$

where $X = \left(\bar{\zeta}_1^{out}, \bar{\zeta}_2^{out}, ..., \bar{\zeta}_q^{out}\right)$. Clearly, just like a regular extreme learning machine ($Y = WX^T$), the output of the Floquet extreme learning scheme involves a simple multiplication of matrices without sensitive or complex filters. This can be also viewed as a mere rescaling of the weight matrix of the digital layer.

The relationship between $\phi$ and $\zeta^{in}$ is set to be ($\phi = \gamma \bar{\zeta}^{in}$), where $\bar{\zeta}^{in}$ and $\gamma$ are the mean of input vector and a scaling factor, respectively. The value of $\gamma$ for learning nonlinear functions is 1 and for other tasks is equal to $2\pi$.

For learning nonlinear functions, Abolone dataset, and forecasting chaotic time-series, we used a supervised learning algorithm, linear regression, to train the readout function. The predicted output is compared with the ground truth, and the error is calculated and used to update the weights in the readout network following the linear regression learning rule.

To train the readout network, for classification task-parallel image processing, we used the Python toolkit Keras, which provides a high-level application programming interface to access TensorFlow. A supervised learning algorithm, softmax regression, was used to train the readout network. A softmax function is

used as the activation function of the readout network to calculate the probability corresponding to the different possible outputs. The cost is calculated following a categorical crossentropy. A standard gradient-based optimization method is used to minimize the cost function and train the output network. There are several ways of converting images into one-dimensional representations. For simplicity, we used a flattened version of downsampled images as an output vector.

## Data availability
The datasets containing the raw information for abalone dataset are from (https://archive.ics.uci.edu/ml/datasets/Abalone), Mnist dataset is from (https://www.tensorflow.org/datasets/catalog/mnist), and COVID-19 dataset is from (https://www.kaggle.com/tawsifurrahman/covid19-radiography-database).

## Code availability
The code used for simulation is a standard finite-difference time-domain (FDTD), and all parameters required are presented in Methods. The codes used for ELM and reservoir computing are standard linear and softmax regressions, which can be found at https://scikit-learn.org/stable/modules/linear_model.html and https://scikit-learn.org/stable/modules/generated/sklearn.linear_model.LogisticRegression.html.

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

## Acknowledgements

A.M. and R.F. acknowledge funding from the Swiss National Science Foundation under the Eccellenza grant number 181232.

## Author contributions

A.M. performed the theoretical and numerical simulations, under the supervision of R.F. All authors participated in writing and revising the manuscript.

## Competing interests

The authors declare no competing interests.
