## [Peer Review File · Nature Communications]

REVIEWER COMMENTS

Reviewer #1 (Remarks to the Author):

The paper of Momeni and Fleury describes an original idea, with a potentially high impact, i.e. how nonlinearities (required for computing) can be achieved in an almost-linear optical system, by time-dependent modulation of its properties. The paper has the potential to open a new area in optical computing / processing. There are, however, there are a few points that must be addressed by the authors.

A central point of the paper is that nonlinearities are hard to achieve in physical systems and achieving nonlinearities requires high input power. The authors cite Refs. 33, 36, 37 to support this statement. It seems that the cited references are totally misunderstood and cited in a wrong context. Ref 33 is a paper that establishes a theory of wave-based RNN's, without giving details about physical implementations. Ref. 36 describes a magnonic system where nonlinearities can be achieved with low power. So the given references does not prove the point, actually, they prove the opposite. The authors should restrict their arguments to optical computing (electromagnetic waves). The present paper addresses a challenge of optical computing schemes. However, this challenge is not general to all forms of wave-based computing– there are a lot of nonlinearities in other types of waves, often at low powers. I would consider changing even the title to restrict the study to EM waves.

My most important technical comment is as follows: the purpose of the Floquet entanglement is to get rid of 'external' nonlinearities that are often required in optical computing schemes. However, it seems that mixers are required in the input side and FFT operations at the output side to generate and detect the required entanglement. There are conversions between phase and amplitude and frequency-encoded information. I am not convinced that the complexity of a circuit requiring external nonlinearities does not come back through the back door. I mean, having the mixers at the inputs, filters at the output, is not it already more complicated than a few nonlinear feedback elements here and there? The authors should describe much more clearly in circuit terms, how inputs / outputs are encoded and how large is the overhead they yield. Writing that 'FFT involves a simple multiplication' or 'using a simple external mixer' is just too vague.

Would be also good to see a comment, how the modulation of the permittivity is achieved physically. I can see that it is more straightforward at microwave frequencies, but not sure about the optical domain.

I was surprised that such a very simple scatterer (or reservoir) with only five scattering centers performs that well on all studied tasks. It would be very good to see some comments on the required internal complexity of the wave substrate. Time series prediction (Mackey Glass equation) requires a high memory capacity from the reservoir, which would usually require a complex 'maze' of wave propagation paths.

From the examples it is not always clear how many outputs are read from the networks and what is number of trainable parameters that is optimized. Since the output weights are digital, it would be useful to state this number. This way to reader could judge whether the heavy lifting is done by the output neural layer or by the internal complexity of the scattering medium.

Optical reservoir computing has quite a large literature. So for at least one of the examples, the authors could provide a benchmark, a quantitative comparison in terms of device complexity and computing performance.

Using externally modulated signals to achieve signal entanglement is a novel idea in optical computing, but similar ideas are explored in oscillator-based neuromorphic computing: see, for example, Vodenicarevic, D., Locatelli, N., Abreu Araujo, F. et al. A Nanotechnology-Ready Computing Scheme based on a Weakly Coupled Oscillator Network. *Sci Rep* 7, 44772 (2017). <https://doi.org/10.1038/srep44772>. I would encourage the authors to explore connections to these fields.

Overall, I think the paper could have significant impact, and the above issues could be addressed by a major revision of the manuscript.

Reviewer #3 (Remarks to the Author):

The manuscript proposes application of phase-shifted time-modulated surfaces to a computing system. I did not find the paper and the novelty of the proposed idea suitable for *Nature Communications*. The paper might be suitable for *Physical Review* journals, and in particular for *Physical Review A*. The phase-shifted time-modulated surfaces were previously proposed in several articles, e.g., [R1]-[R3]. However, the authors have not mentioned previous studies on such surfaces. I suggest to do a careful study on previous phase-shifted time-modulated surfaces.

In addition, the manuscript suffers from several self-citations to the articles that are not directly relevant to the discussed idea, e.g., Refs. [7],[8],[9],[10], [11],[12],[22],[23].

Furthermore, the manuscript does not provide an experimental demonstration of the proposed platform. It would be good if the authors added an experiment.

Overall, the manuscript seems incomplete. The English of the paper is generally poor. The paper needs careful English grammar check. For instance:

1) Page 13: "is a extremely difficult task", "a" should change to "an".

2) Conclusion: "tackled by slower, sophisticated, digital deep neural networks". one should add an "and" before "digital".

Additionally, variables should be in the italic format, e.g., *m* and *n* in page 6 two lines after Eq. (7).

[R1] <https://doi.org/10.1063/1.4939915>

[R2] <https://doi.org/10.1103/PhysRevApplied.12.054008>

[R3] <https://doi.org/10.1103/PhysRevApplied.14.014027>

Response letter, manuscript

We would like to thank all of the reviewers for their time and careful review of our manuscript. In the following, we provide a detailed response to each of the Reviewer's comments and point out all the associated changes. We are confident that the Reviewers will find the revised version of our manuscript considerably improved, and that they will be able to express a positive recommendation for publication.

Reviewer 1

The paper of Momeni and Fleury describes an original idea, with a potentially high impact, i.e. how nonlinearities (required for computing) can be achieved in an almost-linear optical system, by time-dependent modulation of its properties. The paper has the potential to open a new area in optical computing/ processing. There are, however, there are a few points that must be addressed by the authors.

Response: Many thanks for your time and very useful comments on our work, that greatly helped us to improve it and clarify some important points. The revisions are marked by the red color in the revised version of paper.

Comment A1: A central point of the paper is that nonlinearities are hard to achieve in physical systems and achieving nonlinearities requires high input power. The authors cite Refs. 33, 36, 37 to support this statement. It seems that the cited references are totally misunderstood and cited in a wrong context. Ref 33 is a paper that establishes a theory of wave-based RNN's, without giving details about physical implementations. Ref. 36 describes a magnonic system where nonlinearities can be achieved with low power. So the given references does not prove the point, actually, they prove the opposite. The authors should restrict their arguments to optical computing (electromagnetic waves). The present paper addresses a challenge of optical computing schemes. However, this challenge is not general to all forms of wave-based computing- there are a lot of nonlinearities in other types of waves, often at low powers. I would consider changing even the title to restrict the study to EM waves.

Response and revision: We agree with the Referee. We have followed the recommendation of the Referee and modified the title as well as the introduction, including the citation of relevant references.

Comment A2: the purpose of the Floquet entanglement is to get rid of 'external' nonlinearities that are often required in optical computing schemes. However, it seems that mixers are required in the input side and FFT operations at the output side to generate and detect the required entanglement. There are conversions between phase and amplitude and frequency-encoded information. I am not convinced that the complexity of a circuit requiring external nonlinearities does not come back through the back door. I mean, having the mixers at the inputs, filters at the output, is not it already more complicated than a few nonlinear feedback elements here and there? The authors should describe much more clearly in circuit terms, how inputs / outputs are encoded and how large is the overhead they yield. Writing that 'FFT involves a simple multiplication' or 'using a simple external mixer' is just too vague.

Response: There is actually no hidden additional complexity with our scheme, and we apologize if the descriptions made in the initial version of the paper were not detailed enough. To clearly evidence this, we address separately the two points mentioned by the Referee, namely the potentially problematic operations of FFT at the output side and mixing at the input:

- 1) FFT: Even though we work with the amplitude of a few frequency harmonics, the proposed system does not require any filtering operation at the output. Indeed, the FFT is a simple matrix multiplication, just like the linear regression of the digital layer. Both operations are thus performed at the same time, *by rescaling the weights of the digital layer*. Thus, the output layer of our time-Floquet extreme learning machine is not more complex than the one of a regular extreme learning machine: it is still a simple linear regression.

To see this explicitly, consider $x^q(t)$ as the temporal signal received at the output antenna q , and its discrete Fourier transform of the discretized signal $\{x^q(0), x^q(t_0), \dots, x^q((n-1)t_0)\}$ defined by $y_{f_j} = \sum_{k=0}^{n-1} x^q(t_k) \exp(\frac{-2\pi i}{n} jk)$, where $j = 0, \dots, n-1$, and t_0 is the sampling time. The relation between the Fourier coefficients y_{f_j} and the discretized signal is described by the so-called Vandermonde matrix:

$$\begin{pmatrix} y_{f_0} \\ y_{f_1} \\ \dots \\ y_{f_{n-1}} \end{pmatrix} = \begin{pmatrix} 1 & \dots & 1 \\ 1 & \dots & \xi^{n-1} \\ \vdots & \ddots & \vdots \\ 1 & \dots & \xi^{(n-1)^2} \end{pmatrix} \begin{pmatrix} x^q(0) \\ x^q(t_0) \\ \dots \\ x^q((n-1)t_0) \end{pmatrix}$$

where $\xi = e^{-\frac{2\pi i}{n}}$. Generally, we are only interested in the middle harmonic whose component q can be calculated by the scalar product $y_{f_j}^q = F_j X_t^q$, where $F_j = (1 \ \xi^j \ \xi^{2j} \ \dots \ \xi^{j(n-1)})$, $X_t^q = (x^q(0) \ \dots \ x^q((n-1)t_0))^T$, $(\cdot)^T$ is the transpose operation, and j corresponds to desired harmonics.

On the other hand, the single digital layer present in any extreme learning machine use a simple linear regression (which is commonly defined by a linear matrix operation of the form $Y = WX^T$, where W is the weights matrix). Therefore, in our time-Floquet system, we must compose both operations, multiplying the F_j and weight matrices:

$$Y = W(F_j X)^T$$

where $X = (X_t^1, X_t^2, \dots, X_t^q)$. Clearly, just like a regular extreme learning machine ($Y = WX^T$), the output of the Floquet extreme learning scheme involves a simple multiplication of matrices without sensitive or complex filters. This can be also viewed as a rescaling of the weight matrix of the digital layer. In conclusion, the output side of our system is not more complicated than a regular extreme learning machine.

2) Input side:

We encode the input information with simple circuit elements, namely variable attenuators and a voltage-controlled phase shifter. These devices are set externally to a certain operating point once the input is selected, and do not change as the neural network processes a given input. Therefore, no dynamic tuning, or conversion between amplitude and phase, is needed between the temporal signal sent by the generator and the modulation signal: just like in standard learning machines, they are just set externally once an input is selected to be processed by the system.

More details can be found in Figure R1. The amplitude-coding scheme that we employ is simple and physically feasible by using variable attenuators (VOA). Since the values of input vector ξ_n^m are normalized between zero and one, the input information can be encoded on the amplitude of the temporal signal generated by a single signal generator (SG), as illustrated in the figure below. Variable attenuators provide the input node amplitudes depending on the external voltages applied. In addition, a lower frequency oscillator with a voltage-controlled phase shifter (VCP) is used to drive the modulation of the time-Floquet layer. VCPs are tunable, and the applied voltage is simply calculated by the external user from the input vector.

Figure R1: A detailed schematic of the proposed wave-based ELM architecture. On the left, the input side includes a signal generator (SG) with variable attenuators (VOA) that encode the information. A Voltage-controlled phase shifter (VCP) is used to change the phase of the modulation depending on the voltage applied, which is fixed for each input and does not change in time. AA denotes the array of source antennas. V_p is the required voltage to control the VCP ($V_p = \gamma \bar{\xi}^m$), where $\bar{\xi}^m$ and γ are the mean of input vector and a scaling factor, respectively.

Also, in order to comment on the possibility of “using a few nonlinear feedback elements here and there”, we considered some common nonlinear elements such as sensors or detectors with square-law (x^2) nonlinearity. We found that this fails at interpolating highly nonlinear functions (more details are in the response to *Comment A4*). As a result, the proposed computing system is nontrivial, yet simple to implement, and much more effective than more obvious schemes.

Revision: We added all aforementioned explanations in the revised paper, together with the new figure R1 (pages 18 and 19).

Comment A3: Would be also good to see a comment, how the modulation of the permittivity is achieved physically. I can see that it is more straightforward at microwave frequencies, but not sure about the optical domain.

Response : We are happy to clarify this point. Note that the modulation frequency is small compared to the operating frequencies f_1 and f_2 , since $f_m = |\frac{f_1 - f_2}{2}|$, and the modulation depth does not have to be large, as long as one can detect the Floquet harmonics above the noise level. This flexibility allows the proposed time-Floquet layer to be implemented in different frequency ranges.

Figure R2: Realistic physical platforms to implement the time-Floquet layer at (a) Microwave frequencies; (b), in the terahertz range; and (c), in optics.

At microwave frequencies, one can leverage a metasurface that incorporates a single temporally modulated capacitive layer backed by a dielectric layer. As a simple example, consider periodically arranged square patches (or other form of footprint patterns) with varactors soldered between the neighboring patches (see figure R2 (a)). The time-varying modulation $C(t) = C_0(1 - \delta \sin(\omega_m t + \phi))$ is introduced by applying a time-varying voltage to the varactors [R1-R4].

In the terahertz and mid-infrared band, graphene is a good candidate to implement time-varying components due to its tunable electrical conductivity and compatibility with common micro-fabrication technologies. The sheet conductivity of graphene can be effectively modified via electrical bias (see figure R2 (b)). More discussion can be found in the Sec.4 of the supplementary material.

In the optical domain, there are different methods to achieve the needed time-varying responses. One method is to apply a time-varying voltage on special materials such as indium tin oxide (ITO). ITO is one of the most well-known transparent conducting oxide for realization of electro-optical modulators used in telecommunications [R4].

The degenerate doping of ITO can redshift its plasma frequency, leading to a largely tunable optical response at infrared frequencies. Similar to graphene, here, by applying a time-varying voltage, dynamic phase modulation in reflection and transmission can be achieved [R5].

R1: Zhang, L., Chen, X. Q., Liu, S., Zhang, Q., Zhao, J., Dai, J. Y., ... & Cui, T. J. (2018). Space-time-coding digital metasurfaces. *Nature communications*, 9(1), 1-11.

R2: Sounas, D. L., & Alu, A. (2017). Non-reciprocal photonics based on time modulation. *Nature Photonics*, 11(12), 774-783.

R3: Zhang, L., Chen, X. Q., Shao, R. W., Dai, J. Y., Cheng, Q., Castaldi, G., ... & Cui, T. J. (2019). Breaking reciprocity with space-time-coding digital metasurfaces. *Advanced Materials*, 31(41), 1904069.

R4: Wuttig, M., Bhaskaran, H., & Taubner, T. (2017). Phase-change materials for non-volatile photonic applications. *Nature Photonics*, 11(8), 465-476.

R5: Barati Sedeh, H., Salary, M. M., & Mosallaei, H. (2020). Topological space-time photonic transitions in angular-momentum-biased metasurfaces. *Advanced Optical Materials*, 8(11), 2000075.

Revision: We added a section in the Supplementary material (Sec. 4) to discuss realistic physical platforms for the time-Floquet layer in different frequency ranges.

Comment A4: I was surprised that such a very simple scatterer (or reservoir) with only five scattering centers performs that well on all studied tasks. It would be very good to see some comments on the required internal complexity of the wave substrate.

Response: We are happy to clarify this central point of our work: time-Floquet learning machines do not require complex reservoirs because the non-linear mapping produced is much more efficient than what can be achieved by standard non-linearity forms, even of large magnitude. We have generated new results and included a new figure to prove this point and explain it better.

Before going into details of the learning principle, let us remind that ELMs always include two subparts: (i) a fixed efficient nonlinear mapping; and (ii) a simple, quick-to-train, decision layer (linear regression). In the first subpart, it is necessary to make sure that the encoded data maps in a non-sparse way to the feature space, which can be done by mixing efficiently the modal degrees of freedom using scatterers, as mentioned by the Referee. Importantly, we need enough degrees of freedom (modes) and a good coupling between them, so that a modification at each input efficiently affects all outputs.

In a linear system, adding more scattering objects simply change the linear mapping. Yet, when weak non-linearity is considered in the scattering process, increasing multiple scattering is typically a good thing, as it enhances the non-linear character of the mapping. Here, however, adding scatterers does not have a large impact on the performance, because the form of non-linearity is not related to the scattering process but to the Floquet dynamics. Therefore, to understand the performance of our system, one must understand why the non-linear Floquet entanglement is so efficient at turning the input data into linearly separable information in the feature space.

We can explain the learning principle of the proposed computing system with well-known kernel methods. Kernel methods use kernels (or basis functions) to map the input data into a feature space. After this mapping, simple models can be trained on the new feature space, instead of the input space, which can result in an increase in the performance of the models. We can describe the projections of input samples in the feature space by $H = R_n(p(\zeta_{in}))$, where p is the encoding function, here for example $p(\zeta_{in}) = \zeta_{in}(\sin(\omega_1 t) + \sin(\omega_2 t))$, and R_n is a nonlinear and complex function associated with the time-Floquet entanglement. Essentially, R_n can be seen as a kernel function which contains both the multiple scattering occurring in the media and the complex nonlinearity form. This kernel contains several polynomial basis functions, $\{x, x^2, x^3, x^4, \dots\}$ (we can theoretically show it by Taylor expansion of equation 6 in the main paper). Hence, we have a combination of different orders of polynomial mappings with random coefficients in the feature space for each read-out node, resulting in transferring different features of the input data into the feature space. For this reason, the proposed kernel is very efficient in performing all tasks, even when compared with a strongly non-linear Kernel such as the modulus square operator (x^2), typically found in sensors and detectors used in prior arts.

To prove this quantitatively, we compare the feature space projection of our kernel with the form of nonlinearity that is most commonly used in prior arts: a square-law at the detector, x^2 . As a reference, we also look at the purely linear case. As an example, we consider again the nonlinear interpolation of $y=\text{sinc}(x)$. In order to perform well, the data projected in the feature space should be highly nonlinear with respect to the feature coordinates. To visualize this, we use principal component analysis (PCA) to reduce the dimension of the output data, because it lives in a high-dimensional space (10 dimensions, set by the number of readout nodes). PCA is a kind of linear projection that consists in transforming correlated variables into new variables, decorrelated from each other. These new variables are called “principal components” or principal axes. In figure R3a and b, we calculate and plot this projected data in a 3D PCA space for three distinct cases: linear, x^2 nonlinearity, and time-Floquet entanglement. Panels (a) and (b) show that in both the linear and x^2 cases, the projected data is on a line, whereas in the case of the time-Floquet entanglement, the data follows a highly non-linear relation with respect to the principal axes. For this reason, the *sinc* interpolation fails when using both a linear system or one with x^2 non-linearity at the detector (see figure R3 (c) and (d)). Conversely, time-Floquet entanglement is extremely good at performing the sinc problem, because it results in nonlinearity in this high-dimensional space. This explains and demonstrates the crucial role of time-Floquet entanglement in the non-linear mapping and in the high performance of the learning machine.

Figure R3: **PCA analysis of the proposed optical kernel.** (a) and (b) Two different perspectives of projected data for three different cases: i) linear case, ii) square-law nonlinearity (x^2), and iii) the proposed nonlinearity. (c), (d), and (e) sinc(x) interpolation results for three aforementioned cases, respectively.

Revision: We added a figure to the paper to demonstrate the crucial role of time-Floquet entanglement in the non-linear mapping of the extreme learning machine. We also added new explanations and results to the revised paper (pages 10 and 11).

Comment A5: Time series prediction (Mackey Glass equation) requires a high memory capacity from the reservoir, which would usually require a complex ‘maze’ of wave propagation paths.

Response: We agree that predicting time-series requires a memory. In that example, the memory is implemented using a feedback-loop between the output and the input, using the intensity of harmonic waves as reservoir states (see equation (9) in the main paper). This architecture is compatible with the autonomous forecasting of chaotic time-series performed as the test phase. We did not add any new remark concerning this point, since it is already very explicit in the paper. We included this example of time-series forecasting to show that time-Floquet entanglement is also relevant in memory computing.

Comment A6: From the examples it is not always clear how many outputs are read from the networks and what is number of trainable parameters that is optimized. Since the output weights are digital, it would be useful to state this number. This way to reader could judge whether the heavy lifting is done by the output neural layer or by the internal complexity of the scattering medium.

Response and revision: As the reviewer suggests, we mentioned them in the revised version of paper (pages 9, 12 and 16). This data is repeated in the table below, for your information.

Tasks	Number of input nodes	Number of read-out nodes
Learning highly nonlinear functions	10	20
Abalone dataset	8	50
Parallel image classifications	100	100
Forecasting the chaotic Mackey Glass time series	100	50

Comment A7: Optical reservoir computing has quite a large literature. So for at least one of the examples, the authors could provide a benchmark, a quantitative comparison in terms of device complexity and computing performance.

Response and revised: Thank you for the suggestion. We added a section in Supplementary material (Sec. 3) to compare the computing performance of the proposed system with other computing systems for all tasks. This is repeated below for your information:

Tasks	Benchmark	Details
Learning highly nonlinear function (sinc(x))	Ref [31]: RMSE: 0.0039 Our work: RMSE: 0.0015	 Ref [31] used a GRIN 50/125 multimode fiber (MMF) that supports 240 spatial modes. In this work, the authors used a high-power optical pulse in order to excite the nonlinearity of GRIN MMF (input optical peak power pulse equal to 3.43kW).
Abalone dataset	Ref [31]: RMSE:0.126 Our work: RMSE: 0.064	 We used only 20 and 50 readout nodes for interpolating nonlinear functions and the Abalone dataset, respectively, with no power constraint.
Parallel image classifications	Ref [50]: Test accuracy for Mnist-dataset: 79.2% Ref [31]: Test accuracy for Covid-dataset: 83.2% Our work: Test accuracy for Mnist-dataset: 85.3% Test accuracy for Covid-dataset: 88.2%	 The Ref [50] and [31] used all image pixels (for example 28*28 for MNIST dataset) to encode input data. We perform parallel training after down-sampling of input images (for example 10*10 for MNIST dataset). Yet, the classification-accuracy results remain comparable.
Forecasting the chaotic Mackey Glass time series	Ref [49]: Autonomous forecasting: for 50 time-steps Our work: Autonomous forecasting: for 60 time-steps	 Ref [49] employed multiple tungsten oxide (WOx) memristors as a reservoir computing system. In this work, the authors used 20 devices (memristors) and 50 virtual nodes (total of 1000 nodes) to Forecast the chaotic Mackey Glass time series. We used only 100 and 50 input and readout nodes, respectively.

Comment A8: Using externally modulated signals to achieve signal entanglement is a novel idea in optical computing, but similar ideas are explored in oscillator-based neuromorphic computing: see, for example, Vodenicarevic, D., Locatelli, N., Abreu Araujo, F. et al. A Nanotechnology-Ready Computing Scheme based on a Weakly Coupled Oscillator Network. Sci Rep 7, 44772 (2017). <https://doi.org/10.1038/srep44772>. I would encourage the authors to explore connections to these fields.

Response: Thank you for this comment. We think, however, that our work differs from the reference provided by the Reviewer, and that our paper is still largely different from prior art in neuromorphic computing.

This work investigated coupled oscillators based on the Kuramoto model in order to perform pattern recognition. The Kuramoto model captures the dynamics of an oscillator network (we copied the equation below). This equation describes the evolution of an oscillator's phase θ_i as a function of its intrinsic frequency f_i^0 and the influence of the

other oscillators $\sum_j k_{ij} \sin(\theta_j - \theta_i)$. The coupling from oscillator j to oscillator i is modeled through the coupling strength k_{ij} :

$$\frac{1}{2\pi} \frac{d\theta_i}{dt} = f_i^0 + \sum_j k_{ij} \sin(\theta_j - \theta_i) + \text{Noise}$$

In this work, the system of oscillators is intrinsically nonlinear. As clear from this equation, the nonlinearity between $\frac{d\theta_i}{dt}$ and θ_i stems from the sinus function. In order to accomplish pattern recognition, information is encoded directly on f_i^0 and θ_i and *no external modulation* is needed because this equation already provides the required nonlinearity. However, in our work, the wave equation is literally linear even after considering the time-Floquet layer. For this reason, we add a time-Floquet entanglement in order to provide the missing nonlinearity. One can find several nonlinear systems that were recently used for reservoir computing and neuromorphic engineering such as memristor devices. The concept in our paper is literally different, and leads to drastically higher non-linear mappings, as we demonstrated above.

Revision: A brief comparison has been added to the introduction of the revised manuscript.

Comment A9: Overall, I think the paper could have significant impact, and the above issues could be addressed by a major revision of the manuscript.

Response: Thank you for the positive words. We have taken this opportunity to improve the clarity and presentation of our results, and are confident that the revised manuscript has been positively amended and improved.

Reviewer 3

Comment B1: The manuscript proposes application of phase-shifted time-modulated surfaces to a computing system. I did not find the paper and the novelty of the proposed idea suitable for Nature Communications. The paper might be suitable for Physical Review journals, and in particular for Physical Review A.

Response: We respectfully disagree with the evaluation of the Reviewer. Our work is very novel, and as acknowledged by the first Referee, "could have significant impact". The opinion of the Referee is not supported by scientific arguments, but rather by:

- (i) comparisons with papers in a different field, namely time-modulated metasurfaces, none of which deal with neuromorphic computing
- (ii) the lack of experiment, which is not a scientifically receivable criticism on its own. The history of science proves that many theory papers proposing new paradigms can have large impact and lead to major breakthrough in a variety of fields.

The topic of neuromorphic engineering, ELM and RC processing is of interest to many research communities, and proposing an efficient wave-based learning machine is still an unsolved challenge especially in optics. Several works in this journal have been published to improve the computing performance of reservoir computers with different architectures (see for example Ref [49-55]). In terms of novelty, our work has made four key innovations that advance significantly the state-of-the-art in this topic:

- 1) Solving the vexing challenge of high-power requirements of standard nonlinear schemes by leveraging the time-Floquet entanglement technique (**modulation phase or depth**).
- 2) Increasing the reservoir size of wave-based reservoir computing using both spatial and spectral domains in order to reduce training error values (benchmarks) without imposing additional filters nor a larger computational overhead.
- 3) Possibility of parallel training of several datasets by leveraging the spectral domain (several frequency bands).
- 4) The proposed kernel, which maps the input data to the feature space, is reconfigurable and very efficient in performing well all tasks, even compared with standard methods like square-law at the detector (x^2) (see pages 10 and 11 of the revised paper).

Revision: We modified the introduction section of the paper in order to stress better on the novelties of this work. We added a new section to the supplementary material of the paper to show that time-Floquet entanglement is not only related to phase modulation. We can indeed also leverage a dynamical modulation depth (δ_m) in a Floquet-layer without involving the modulation phase. The revisions are marked by the **magenta** color in the revised version of paper.

Comment B2: The phase-shifted time-modulated surfaces were previously proposed in several articles, e.g., [R1]-[R3]. However, the authors have not mentioned previous studies on such surfaces. I suggest to do a careful study on previous phase-shifted time-modulated surfaces.

[R1] <https://doi.org/10.1063/1.4939915>

[R2] <https://doi.org/10.1103/PhysRevApplied.12.054008>

[R3] <https://doi.org/10.1103/PhysRevApplied.14.014027>

Response:

- It is important to stress that the focus of this study is not phase-shifted time-modulated surfaces or their applications, such as nonreciprocity or other wave manipulations, that have been well investigated previously, as mentioned by the Referee. The topic of this paper is neuromorphic engineering and wave-based reservoir computing. Any opinion about the impact and novelty of our work should therefore be supported by comparing it with the state-of-the-art in this field. In short, we propose a novel wave-based extreme deep-learning machine that gets rid of the usual high-power requirements required to induce nonlinearity, and without a time-consuming training process. In comparison to prior works, the proposed power-efficient wave-based extreme deep-learning machine, compatible with both ELM and RC architectures, is reconfigurable and has an ability of parallel training (training several datasets at the same time, section 2.3 in the main text). **A detailed comparison with prior art using quantitative metrics clearly demonstrates the potential of our method (see the table of Comment A7).**
- Secondly, we show the flexibility of the proposed neuromorphic computing system by leveraging the **dynamical modulation depth** (δ_m) of the Floquet-layer without involving the modulation phase. We added a section in the *Supplementary material (Sec. 2)* in order to show this possibility without involving the modulation phase. In this case, input information is encoded in **modulation depth** of Floquet-layer in order to induce the required nonlinear entanglement, and no phase shifters are needed. In Sec. 2 of the Supplementary material, we shown the efficiency of this approach using kernel method and PCA analysis. **In this context, the topic has literally nothing to do with the field of phase-shifted time-modulated metasurfaces.** The main idea to reach high nonlinearity by time-Floquet entanglement of the input information, which is a unique idea in wave-based reservoir computing.

Figure R4: PCA analysis of the proposed optical kernel for dynamical depth modulation case. (a) and (b) Two different perspectives of projected data for three different cases: i) linear case, ii) square-law nonlinearity (x^2), and iii) the proposed nonlinearity. (c), (d), and (e) sinc(x) interpolation results for three aforementioned cases, respectively.

Revision: We added a section in supplementary material (Sec. 2). Although we cited a few such papers when we introduced time-Floquet systems (Ref [33-41] in the main paper) in the introduction section, we added the ones proposed by the Referee, and a few other relevant works in the introduction section.

Comment B3: The manuscript suffers from several self-citations to the articles that are not directly relevant to the discussed idea, e.g., Refs. [7],[8],[9],[10], [11],[12],[22],[23].

Response and revision: We have now significantly reduced the number of self-citations.

Comment B4: The manuscript does not provide an experimental demonstration of the proposed platform. It would be good if the authors added an experiment.

Response and revision: We proposed a novel paradigm in the area of wave-based deep-learning computing systems, which is fully supported by both theoretical and numerical simulations. Although we agree with the referee in the fact that experimental results may certainly enrich the presented study, the research topic is very dynamic and still within the early stages of its development, in which theoretical and design advances are particularly important. Thus, accomplishing measurements goes way beyond the scope of this paper and may be the purpose of a future study. There are also many papers in the field of wave-based analog computing in which the achievements are validated through numerical/theoretical results only, without restricting their impact. We note that the feasibility of a uniform, slow, and weak modulation scheme does not constitute a particular experimental challenge, as detailed in *comment A3*.

Comment B5: The English of the paper is generally poor. The paper needs careful English grammar check. For instance:

- 1) Page 13: "is a extremely difficult task", "a" should change to "an".
- 2) Conclusion: "tackled by slower, sophisticated, digital deep neural networks". one should add an "and" before "digital".
- 2) Additionally, variables should be in the italic format, e.g., m and n in page 6 two lines after Eq. (7).

Response and revision: We have carefully proofread the manuscript, corrected all grammar errors, and improved the style in general.

* * *

REVIEWERS' COMMENTS

Reviewer #1 (Remarks to the Author):

The authors have done an exemplary job in revising the paper, adding new simulations and a significant number of details, clarifications and supplementary sections. My questions and concerns are answered now.

Response letter, manuscript

We would like to thank all of the reviewers for their time and careful review of our manuscript.

Reviewer #1

The authors have done an exemplary job in revising the paper, adding new simulations and a significant number of details, clarifications and supplementary sections. My questions and concerns are answered now.

Response: Thank you very much for the valuable time you have spent reviewing our manuscript and providing insightful comments to help significantly improve the quality of our work. We are glad to see that you are satisfied with our revision.

* * *